# Exploring Temperature-Related Effects in Catch Crop Net N Mineralization Outside of First-Order Kinetics

Jorge Federico Miranda-Vélez * and Iris Vogeler

Department of Agroecology, Section for Soil Fertility, Aarhus University Foulum, 8830 Tjele, Denmark; iris.vogeler@agro.au.dk
* Correspondence: jorge_mv@agro.au.dk

**Abstract:** Catch crops are an effective method for reducing nitrogen (N) leaching in agriculture, but the mineralization of incorporated catch crop residue N is difficult to predict and model. We conducted a five-month incubation experiment using fresh residue from three catch crops (hairy vetch, fodder radish and ryegrass) with three temperature treatments (2 °C, 15 °C and 2–15 °C variable temperature) and two termination methods (glyphosate and untreated). Mineral N (ammonium and nitrate) in soil was quantified at 0, 1, 2, 4, 8 and 20 weeks of incubation. Ammonium accumulation from residue decomposition showed a lag at low and variable temperature, but subsequent nitrification of the ammonium did not. Mineral N accumulation over time changed from exponential to sigmoidal mode at low and variable temperature. Incubation temperature significantly affected mineralization rates in a first-order kinetics (FOK) model, while plant type and termination method did not. Plant type alone had a significant effect on the final mineralized fraction of added catch crop N. FOK models modified to accommodate an initial lag were fitted to the incubation results and produced better goodness-of-fit statistics than simple FOK. We suggest that initial lags in residue decomposition should be investigated for the benefit of mineralization predictions in cropping models.

**Keywords:** nitrogen mineralization; catch crops; incubation; soil organic matter; glyphosate; soil microbes; first-order kinetics; modeling

---

## 1. Introduction

In agriculture, catch crops are a widespread and effective method for reducing postharvest soil mineral nitrogen and thereby leaching of nitrogen after harvest and during the autumn-winter season [1–3]. However, nitrogen taken up by the catch crops is subject to mobilization after the catch crops are terminated and incorporated into the soil, and their residues are degraded by soil microbes [4].

The degree and rate of plant residue decomposition in agricultural soils are critical factors in determining the fate of the mobilized catch crop nitrogen. Ideally, this nitrogen (N) is taken up by the subsequent main crop, but in humid temperate climates with high precipitation, there is a risk of leaching mobilized catch crop nitrogen [5]. Apart from leaching, mobilized soil nitrogen can also be lost via nitrous oxide emissions from both nitrification and denitrification due to the creation of low-oxygen hotspots around the incorporated plant residues [6] and the introduction of fresh labile carbon sources [7]. The network of microbial processes that results in net plant residue nitrogen mobilization is affected by a multitude of environmental conditions, ranging from soil texture [8] and temperature [9] to the characteristics of the plant residue [10], the methods of catch crop termination and incorporation [11] and the management history of the soil [12].

Dynamic models of cropping systems, such as Daisy [13], APSIM [14], HERMES [15,16], DNDC [17] and DSSAT [18] (among many others) are powerful tools for predicting the fate of nitrogen in agricultural soils, including the turnover and transformation of the nitrogen held in terminated catch crops. However, most models capable of simulating

entire cropping systems avoid detailed mechanistic descriptions of the soil microbiome and its interactions with added residue and soil organic matter in favor of "black box" approaches. These lump together the complex network of microbe–substrate–environment interactions in relatively simplified functions. The functions, in turn, are defined with a mixture of empirical and ad hoc parameters, often including microbial biomass as a parameter or pool, but lacking any explicit description of the microbial community, their diversity and how this is related to processes and process rates [19]. A single notable exception is the ORCHIMIC model [20], currently under development with the aim of being embedded in larger cropping systems models.

The most-used description of plant residue or soil organic matter turnover in cropping models is based on first-order kinetic (FOK) processes representing carbon and nitrogen flows between more or less abstract functional fractions (or pools) of the soil organic matter. Each first-order kinetic interaction is defined by a rate constant and a collection of scaling factors that account for the effect of temperature, soil water content, pH and clay content, among others [21]. The functions used for computing these factors are derived more or less empirically, often via fitting soil incubation results to FOK models and examining the effect of different incubation conditions on the overall turnover rate constant [22]. This approach, although robust, has certain limitations, which we propose requires exploring.

In particular, while cropping models based on the FOK approach generally account for the slowing of organic matter turnover at low-temperature, low water content and pH extremes, they inherently assume an instantaneous reaction of decomposer microbial communities to residue incorporation. Microbes and microbial communities, however, are known to take time to adapt before increasing or decreasing their activity after a change in their environment, such as the addition of fresh substrate [23], and require time to fully colonize incorporated plant residues [24]. Furthermore, both of these preambles to residue decomposition are also likely to be slower at low temperatures, as most soil microbial processes are, increasing the delay between substrate addition and observable net N mineralization.

For these reasons, we hypothesize that besides the effects from plant type on overall N mineralization, there must be a transient temperature-dependent effect on net N mineralization rate shortly after plant residue incorporation. Additionally, we raise the question of whether the FOK model is adequate for describing the early accumulation of mineral nitrogen at different temperatures or whether alternative, still simplified, mathematical expressions could be proposed that take into account the effect of temperature on the activation of decomposer soil microbes, as well as the colonization of plant residues.

In the present study, we carried out a five-month incubation experiment at low temperatures typical of temperate autumn-winter seasons (2°C and 15 °C) to investigate net nitrogen mineralization from catch crop residues with a focus on the initial stages of residue decomposition. We selected three common catch crops belonging to three plant families used widely in agriculture (mustards, grasses and legumes) in order to represent a realistic variety of catch crop residue types. The objective was to identify any non-ideal behavior in decomposer soil microbes reflected as changes in the amount or pattern of net N mineralization during incubation independently from plant type. Given that soil microbial decomposer communities take time to adapt after a change in temperature [25], and different groups of soil microbes have been known to respond differently to sudden temperature changes under incubation conditions [26], a variable temperature treatment oscillating between 2 °C and 15 °C was also considered.

Finally, we consider that herbicides, such as glyphosate, are often used for catch crop chemical termination together with or as an alternative to mechanical termination [27,28]. This method of catch crops termination has the potential for affecting the quality of incorporated residues by initiating tissue senescence and possibly degradation days before incorporation into the soil, which in turn has the potential to affect the timing or rate of catch crop residue N mineralization. Indeed, although little information exists on the topic, glyphosate has been observed to enhance catch crop residue carbon and nitrogen

mineralization similarly to mowing when compared to direct incorporation [11]. Since this effect could interact with both temperature and plant type on early N mineralization from catch crops, chemical termination was also included as a treatment for two of the catch crop types in this study.

## 2. Materials and Methods

### 2.1. Soil and Plant Samples

All soil and plant samples were collected from the Catch Crop Screening experimental field at Foulumgaard, Denmark, at the Department of Agroecology, Aarhus University.

In late March 2019, approximately 80 L of soil were taken from the top 20 cm of an experimental plot treated with glyphosate early in October 2018, which had thus remained bare for the 2018–2019 autumn–winter season. Bulk density, texture and soil water content samples were taken from the same plot at a depth of approximately 100 mm using 100 cm$^3$ steel rings.

The collected soil had a mean clay (<2 μm), silt (<200 μm) and sand (<2 mm) content of 6.2% 32.2% and 57.7%, respectively, a mean organic matter content of 3.9% by mass, and a mean C:N ratio of 12.97. The mean soil bulk density from ring samples was 1.26 Mg m$^{-3}$, and the mean gravimetric water content (GWC) at sampling was 21.7% by mass, approximately 95% of field capacity (the GWC of the soil samples at pF = 2.0 was 22.8%, as determined from pressure plate measurements). All soil for incubation was sieved to 5 mm and mixed by piling using a shovel before storage at 2 °C for approximately 2 weeks until sample preparation.

We selected three catch crops from the field, hairy vetch (HV; *Vicia villosa* "Hungvillosa"), ryegrass (Rg; *Lolium perenne* "Mathilde") and fodder radish (FR; *Raphanus sativus* var. Oleiformis "Brutus"), each grown in a separate plot. These species belong to three distinct plant families (legumes, grasses and mustards) and are frequently used as catch crops in conventional agriculture both alone and in mixtures in Denmark [29].

All three catch crop types were sown in late August 2018 and overwintered in the field. In particular, a mild winter from 2018 to 2019 allowed the fodder radish to survive until spring and developed sizeable taproots. However, none of the catch crops had begun flowering at the time of this experiment. In mid-March 2019, half of the surface of the plots containing the hairy vetch and ryegrass catch crops were sprayed with herbicide (Glyphomax HL at a rate of 2–3 L ha$^{-1}$) to produce the chemically terminated hairy vetch (HV + G) and ryegrass (Rg + G), while the other half was left untreated. Two weeks after herbicide application, plants on each half of each plot were clipped as close to the ground as possible without including any soil or root matter. On the same day, whole fodder radish plants were pulled out of the ground, washed, and separated into root, stem and leaves. Small subsamples (approx. 10 g f.w.) of all plant types were taken to determine water content at the time of sampling by drying at 105 °C.

All remaining plant material was stored at 2 °C for one week, after which all plant types were prepared for incubation on the same day.

### 2.2. Plant C:N Analyses

A representative sample of the collected plant material was dried at 60 °C for 48 h and then milled to a fine powder. The roots and stems of the fodder radish were discarded. Quintuplicate representative 1 g subsamples of the dried and milled plant materials were analyzed for total carbon and nitrogen using an Elementar Vario Max Cube organic elemental analyzer. The plant residue used for incubation had a mean water content between 74% and 85% and contained approximately 5–8 mg total nitrogen and 58–108 mg total carbon (Table 1). The water content of the chemically terminated residues was not measured separately and thus was assumed the same as that of their untreated counterparts. However, only mild wilting was observed at the time of collection, and small differences in net added N due to water losses were judged not to factor in the timing of net N mineralization from the chemically terminated residues.

**Table 1.** Mean properties of the plant residue added to the incubation columns.

| Plant Type | Water Content (%) | % C d.w. | % N d.w. | Total C per 1 g f.w. (mg) | Total N per 1 g f.w. (mg) | C:N |
|---|---|---|---|---|---|---|
| HV | 85.3 | 42.58 | 5.26 | 62.55 | 7.73 | 8.09 |
| Rg | 73.7 | 40.93 | 2.29 | 105.20 | 6.02 | 17.87 |
| FR | 85.0 | 39.03 | 4.59 | 58.29 | 6.85 | 8.50 |
| HV + G | 85.3 | 41.13 | 5.07 | 60.42 | 7.45 | 8.11 |
| Rg + G | 73.7 | 40.95 | 2.03 | 107.62 | 5.33 | 20.17 |

*2.3. Incubation Setup*

A full-factorial incubation experiment was conducted with incubation temperature and plant type as experimental treatments. Each plant type (HV, FR, and Rg), including the two that were treated with glyphosate (HV + G and Rg + G), was split between three temperature treatments: 2 °C constant temperature, 15 °C constant temperature, and variable temperature alternating between 2 °C and 15 °C every 2 to 3 days. These temperatures correspond approximately to the maximum and minimum average daily topsoil temperatures typically observed between the months of September and March at the Foulumgård Experimental Station.

Additionally, a control treatment consisting of only soil was included to account for background decomposition of soil organic matter (SOM) at all three temperatures.

The experimental units consisted of 200 $cm^3$ re-packed soil cores (contained in 250 $cm^3$ stainless steel cylinders) to which different types of plant residue were added and which were incubated at different temperatures for 20 weeks.

All treatments were prepared in triplicate to be sampled destructively at six times of incubation (0, 1, 2, 4, 8 and 20 weeks), resulting in a total of 288 experimental units (week 0 was not split between different temperatures, as this would not have had any effect).

*2.4. Incubation Column Preparation*

Preliminary column preparation was similar to that used by [4] consisted of packing the sieved and mixed soil into small (100 $cm^3$) steel cylinders by gradually adding and pressing four portions of 31.5 ± 0.05 g with a set of custom pistons to a final bulk density of 1.26 Mg $m^{-3}$, equal to that measured in the field.

Plant material was then cut into portions 20–30 mm in length, taking care to damage the tissue as little as possible. The entirety of the aboveground material was used for hairy vetch and ryegrass, but only the leaves and petioles were used for fodder radish, excluding the hard stems.

One gram fresh weight of the cut plant material was placed between two small soil cylinders, which were gently pressed together into a larger (250 $cm^3$) cylinder so that bulk density and compaction remained unchanged, but soil volume was doubled. This method is considered analogous to the coarse mixing of plant material by field machinery, were partially shredded plant residues are turned and buried nonhomogeneously between large soil aggregates by the plow. The control samples were similarly prepared by pressing together two smaller cores into a large one without any added plant material.

The large incubation columns were capped with plastic lids with small breathing holes punched into them and placed in 2 °C and 15 °C climate-controlled rooms according to their temperature treatments. The 2–15 °C variable temperature treatment was achieved by moving the corresponding columns between the two rooms at 2–3 day intervals, keeping a schedule that ensured equal total lengths of time at each temperature. The temperature in the sample containers was monitored for the experiment's duration using wireless temperature loggers to account for unexpected temperature variations inside the climate rooms. Due to the high field water content in the soil, no water was added during column preparation. Water content was monitored by weighing each cylinder at 2-week intervals

and adding demineralized water with a syringe as necessary to maintain constant weight through incubation.

### 2.5. Mineral Nitrogen Extraction and Quantification

During destructive sampling, each column's contents were transferred into an aluminum tray, and any remaining discernible plant material was cut into 1–2 mm portions, after which the entire sample was thoroughly mixed.

Mineral nitrogen (nitrate, $NO_3^-$, and ammonium, $NH_4^+$) was then extracted by mixing 25 g representative samples of column material into 100 mL KCL 1 M aqueous solution and shaking in a spinning rack at 20 rpm for 30 min. The resulting extract was filtered using quantitative ashless paper filters previously rinsed with 50 mL of the same KCl solution. The filtrate was immediately stored at $-18\,^{\circ}$C awaiting analysis.

$NO_3^-$ and $NH_4^+$ concentrations in the KCl extract were determined colorimetrically using a Seal Analytical AA500 auto-analyzer with the methods described by Best [30] and Crooke and Simpson [31], respectively, then transformed to the total mass of nitrate-N and ammonium-N ($NO_3$-N and $NH_4^+$-N, in mg-N) in each sample using water content at extraction and sample mass measurements.

### 2.6. Denitrifying Enzyme Activity (DEA) Assay

In order to rule out important nitrogen losses through denitrification during incubation, DEA assays were carried out in subsamples of 20-week, 15 $^{\circ}$C incubation samples from all plant types. These subsamples were created by extracting smaller cores (approx. 1 cm diameter) from the incubation cores and dividing them into the edge (top and bottom quarters of the small core, composited) and middle (two middle quarters of the small core) portions.

The procedure for this assay was the phase 1 acetylene denitrification assay with glucose as substrate, unlabeled potassium nitrate as nitrogen source and chloramphenicol as protein synthesis inhibitor, as carried out by Duan et al. [32] and described by Tiedje et al. [33]. Headspace gas samples were taken at 15, 45, 75, 135 and 195 min of incubation during the assay into evacuated vials previously filled with helium gas. $N_2O$ concentration in the gas samples was measured by gas chromatography in a dual-inlet Agilent 7890 GC-ECD system.

### 2.7. Statistical Analysis

All statistical analyses were carried out in R version 3.5.3 [34]. Firstly, we calculated the net mineralization of residue-bound nitrogen ($N_{net}$ in mg-N) during incubation by subtracting the mean total mineral nitrogen ($NO_3^- + NH_4^+$, in mg-N) in the control samples at each temperature and incubation time from the total mineral nitrogen in corresponding samples containing plant residue, similarly to Thomsen et al. [4]. To compare the effects of treatments after the full course of incubation, we defined the maximum net mineralization ($N_{max}$) as the mean $N_{net}$ at 20 weeks of incubation for each plant type and the total added plant residue N ($N_0$) as 1000 mg fresh plant residue multiplied by water and total nitrogen contents.

We then compared $N_{max}$ values from HV, FR and Rg at all incubation temperatures by constructing linear regression models (R base function *lm*) using $N_{max}/N_0$ as the response variable and plant type (hairy vetch, fodder radish or ryegrass) and temperature treatment (2 $^{\circ}$C constant, 15 $^{\circ}$C constant or 2–15 $^{\circ}$C variable temperature) as predictor factors. Chemically terminated plant types (HV + G and Rg + G) were not considered in this step since this treatment is not expected to affect the final mineralized fraction of added residue N. The full models were reduced by removal of the non-significant ($p > 0.05$) terms based on analyses of variance comparing the maximum-likelihood estimates of nested models upon removal of single terms starting from four-way interactions (R base function *drop1*).

Additionally, we employed a linear regression model to analyze the effect of plant type, incubation temperature and termination method on the rate of increase in $N_{net}$ during

the eight initial weeks of incubation, independently from $N_{max}$. To do this, we applied a transformation based on a first-order kinetics model:

$$N_{net} = N_{max}\left(1 - e^{-Kt}\right),$$ (1)

where the exponential term can be isolated and removed using a natural logarithm:

$$\ln\left(1 - \frac{N_{net}}{N_{max}}\right) = -Kt,$$ (2)

resulting in a linear function between $N_{net}/N_{max}$ and incubation time, with the FOK rate constant $K$ as the slope in units of weeks$^{-1}$.

However, due to random error, some $N_{net}$ values may be larger than $N_{max}$, causing the difference in the left-hand side of Equation (2) to take negative values, for which the logarithm function is not defined. To avoid this issue, both sides of Equation (2) were squared, taking advantage of the power-to-a-power property of exponents on the right-hand side, and ensuring the left-hand side is always positive:

$$\ln\left(\left(1 - \frac{N_{net}}{N_{max}}\right)^2\right) = -2Kt$$ (3)

The transformed $N_{net}$ values were used as a response variable in a zero-intercept linear regression model with incubation time as a continuous predictor and plant type and incubation temperature as predictor factors. Where significant interactions between incubation time and the other predictors were observed, post-hoc multiple comparisons of model estimates were carried out using Sidak corrections for p values (*R* package "emmeans" [35]), and estimates of the regression slopes (the rate constant $K$ in the FOK model) were extracted from the highest-order interaction involving incubation time.

Chemically terminated plant types were tested separately from untreated plant types, focusing on determining any significant interactions between other factors and chemical termination.

### 2.8. Approaches Outside FOK

We proposed three simple alternative models to test whether minimal modifications to the FOK model can be employed to account for initial delays or lags in net N mineralization. The first alternative model (Exp) is an arbitrary modification of the FOK model in the form used to describe net N mineralization (Equation (2)), where it is simply multiplied by another exponential term. This effectively scales $N_{max}$ to a value, which converges asymptotically with time from zero to 1 at a rate determined by a "lag parameter" $L$, which has the same units as the FOK rate constant $K$:

$$N_{net} = N_{max}\left(1 - e^{-Lt}\right)\left(1 - e^{-Kt}\right)$$ (4)

The lag term in this function can be interpreted as the freshly added plant material becoming gradually more available for decomposition with time, for instance, as the surface of the residue becomes gradually colonized by decomposer soil microbes.

The second alternative function (Gom) is a type of Gompertz equation, which has been suggested as an alternative to describe net N mineralization in forest soils by Ellert and Bettany [36]. This function was derived by adding the same exponential term and lag parameter $L$ as in Exp, as a factor in the differential form of the FOK model,

$$\frac{dN}{dt} = -KN\left(1 - e^{-Lt}\right)$$ (5)

which yields a separable first-order differential equation, and thus the regular form of the Gom model can be obtained by integration:

$$N_{net} = N_{max}\left(1 - e^{K\left(\frac{1}{L} - \frac{e^{-Lt}}{L} + t\right)}\right) \qquad (6)$$

While the Exp and Gom models employed an arbitrary exponential function as a lag term, the final function (Mon) was derived employing a type of hyperbolic function sometimes referred to as a Monod-type function. This type of hyperbolic function is often used to empirically model microbial growth rates under substrate-limiting conditions [37], which makes it appropriate to introduce a gradual increase in net N mineralization rates given a limited amount of surface contact to plant residue and a limited supply of labile N. Thus, we added a Monod-type function, parametrized by the dimensionless Monod constant $K_s$, as a term in the differential form of the modified FOK model:

$$\frac{dN}{dt} = -KN\left(\frac{t}{K_s + t}\right), \qquad (7)$$

and integrated the resulting separable differential equation to obtain the regular form of the Mon model:

$$b)\ N_{net} = N_{max}\left(1 - \frac{e^{-Kt}(K_s + t)^{KK_s}}{K_s^{KK_s}}\right) \qquad (8)$$

Equations (4), (6) and (8), as well as Equation (2), were used to fit $N_{net}$ data over the entire incubation period for each combination of plant, temperature and glyphosate treatments using nonlinear least-squares regression with the Levenberg–Marquardt fitting algorithm (R function nlsLM in package "minpack.lm" [38]). The parameter $N_{max}$ in all equations was fixed using $N_{net}$ at 20 weeks of incubation as before, while the parameters $K$, $L$ and $K_s$ (where relevant) were fitted automatically by the nlsLM algorithm. The goodness of fit for each of the equations considered calculated as the root mean square error (RMSE) of the fitted values against the observed net N mineralization (function rmse in *R* package "Metrics" [39]). These goodness-of-fit values were then compared between models for each plant type, temperature and glyphosate treatment combination.

## 3. Results

### 3.1. Mineral N Accumulation in Samples

Total resident ammonium nitrogen ($NH_4^+$-N, Figure 1a) in the samples increased noticeably in all plant treatments at different times following the start of incubation, depending on incubation temperature. Total contents peaked at one week of incubation for all plant residues at 15 °C, at two weeks for all plant residues at 2–15 °C, and at four weeks for all plant residues at 2 °C, falling to nearly zero thereafter before the end of incubation in all treatments. A much smaller peak in total $NH_4^+$-N was present in the control samples and seemed to reach a maximum within the first 2 weeks regardless of temperature treatment, after which total resident ammonium N remained nearly at zero for the remainder of the incubation.

Total resident nitrate ($NO_3^-$-N, Figure 1b) increased monotonically throughout the incubation in all treatments, including controls. The timing in the accumulation of $NO_3^-$-N seems to follow that of $NH_4^+$-N, showing the highest increase in $NO_3^-$-N immediately after peak $NH_4^+$-N content for all plant and temperature treatments, except control samples.

Total mineral nitrogen (N-min, Figure 1c) content in the columns also increased monotonically throughout the incubation for all treatment combinations and controls. For all plant treatments, accumulation of mineral nitrogen seems to not only occur at a lower rate at 2 °C when compared with the higher temperature treatments, but also seems to increase in rate during weeks 0 to 4, reach an inflection point between weeks 4 and 8, and slow down between weeks 8 and 20. Similar behavior was observed in the variable temperature treatment, where the inflection point was reached around week 2 (Figure 1c).

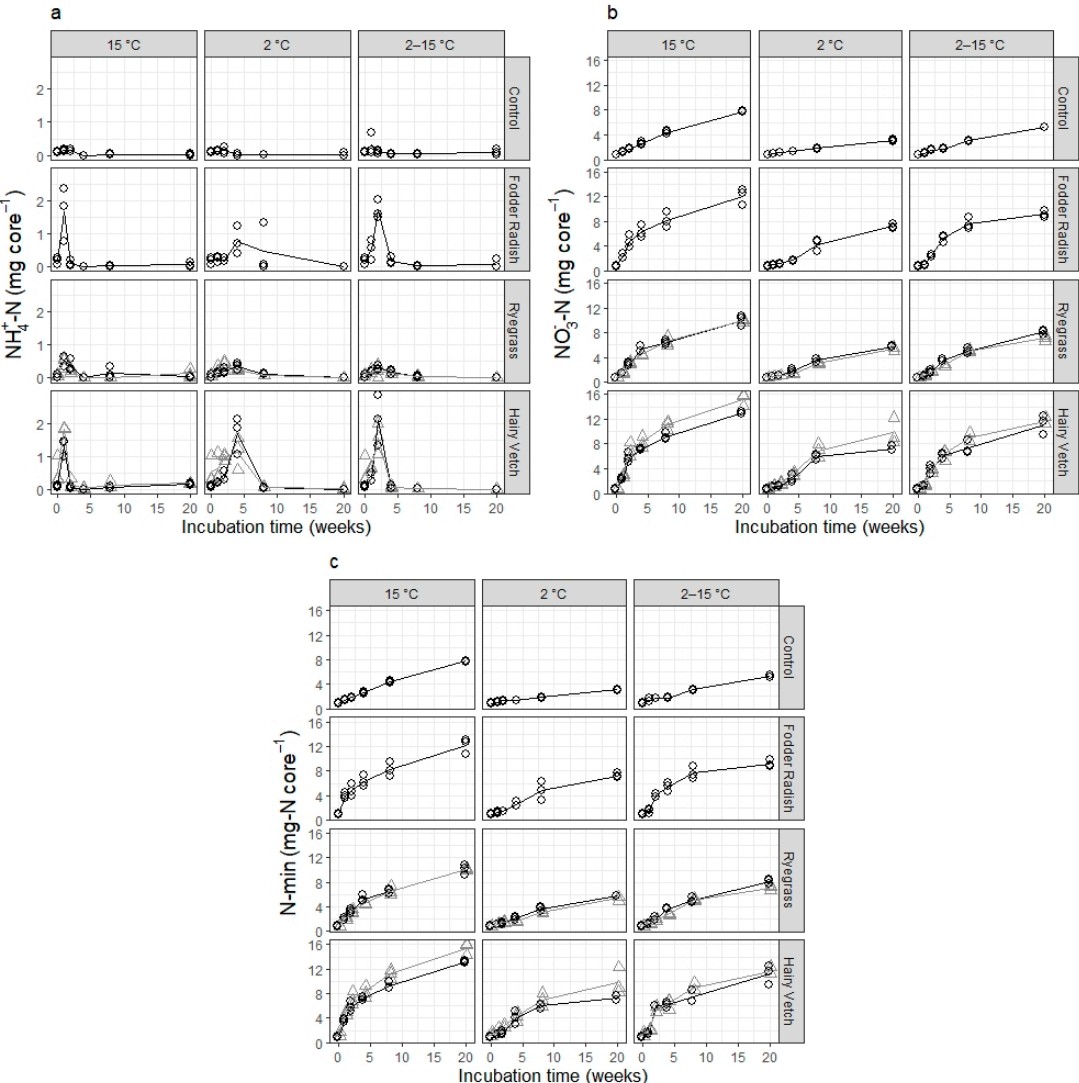

**Figure 1.** (**a**) Ammonium-N, (**b**) nitrate-N and (**c**) total mineral nitrogen in soil cores containing catch crop residue incubated at high (15 °C), low (2 °C) and variable (2–15 °C) temperatures. The gray lines and triangles (△) represent the chemically terminated catch crops, while the black lines and circles (○) represent untreated catch crops. Symbols represent individual measurements, while lines represent mean values. All amounts are reported as total weight (in mg of N) per soil core (~250 mg dry weight).

Background mineralization of soil organic matter (SOM) quantified in the control treatments appears to have made up a sizeable portion of the total net N mineralization, as shown in Figure 1c. Therefore, assuming there is no interaction between SOM and catch crop residue net N mineralization, $N_{net}$ provided a much clearer picture of the rate and timing of net added plant residue N mineralization (Figure 2).

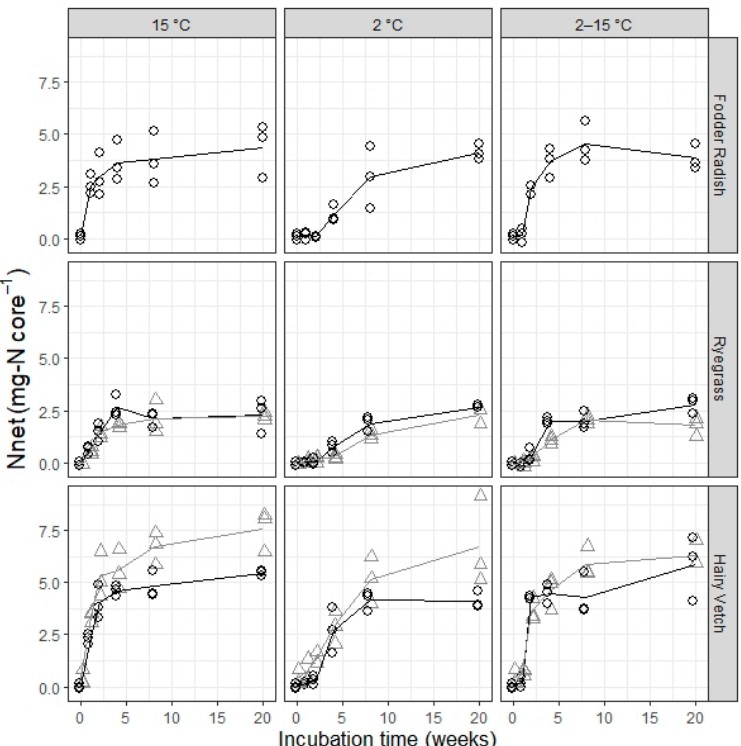

**Figure 2.** Net total N mineralization from catch crop residues with the mean background mineralization removed ($N_{net}$) Black dots (○) and lines represent individual measurements and mean values, respectively, in untreated catch crop types. Gray triangles (△) and lines represent individual measurements and mean values, respectively, in chemically terminated catch crop types. All amounts are reported as total weight (in mg of N) per soil core (~250 mg dry weight).

### 3.2. Denitrifying Enzyme Activity (DEA)

We found no evidence of denitrification from DEA in high-temperature (15 °C) incubation column subsamples after 20 weeks of incubation. This was expected, given that soil moisture was kept slightly under field capacity, and care was taken not to add water in excess during incubation. Furthermore, the incubation cylinders were ventilated, which may have helped to prevent low-oxygen conditions in the soil. Finally, resident nitrate contents in the samples remained relatively low (under 15 mg $NO_3^-$ N in approximately 250 g of soil) throughout incubation, further reducing the likelihood of denitrification.

### 3.3. Final Net N Mineralization

The temperature did not significantly influence the net fraction of added residue N mineralized after 20 weeks of incubation ($N_{max}/N_0$) for any plant type ($F_{2,22} = 0.01$, $p = 0.462$, Figure 3), and, while there seems to be a decrease in $N_{max}/N_0$ with a low temperature in HV, no significant interactions were observed between plant type and temperature treatments.

$N_{max}/N_0$ reached an average of 60%, 42.9% and 66.5% for FR, Rg and HV, respectively, when considering all incubation temperatures together, where the difference was significant between ryegrass and each of the other two plant types, but not between HV and FR.

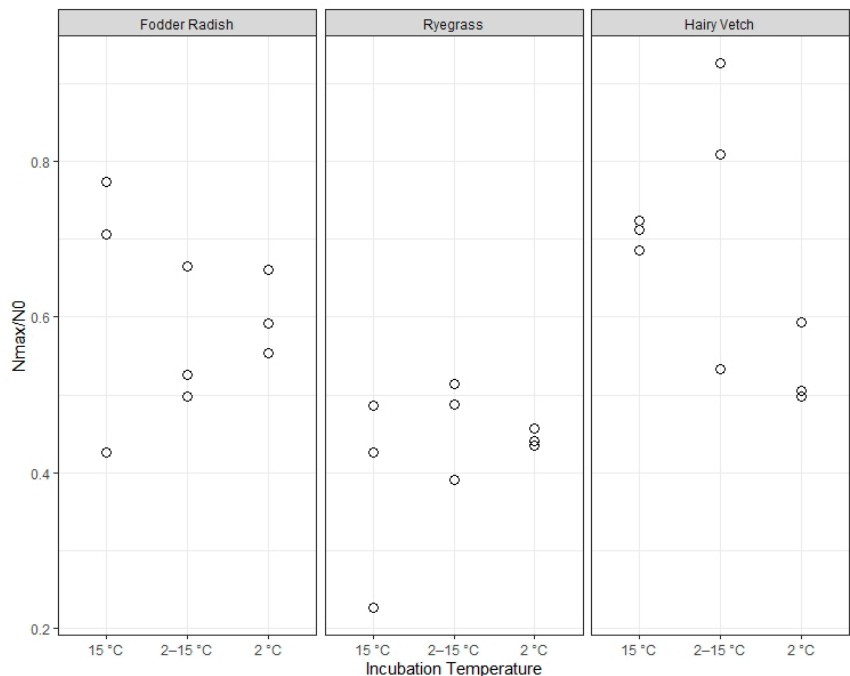

**Figure 3.** Final net plant residue N mineralization as fractions of total added residue N ($N_{max}/N_0$). Dots (○) represent individual soil core samples, incubated in triplicates at high and low constant temperatures as well as variable temperature, with three types of catch crop residue. Mean background net N mineralization from control samples (no plant residue) was subtracted.

### 3.4. Overall Plant N FOK Turnover Rate Constants

Linear regression using the transformation shown in Equation (8) on data from untreated catch crop residue showed no significant main effects from plant type, but a significant three-way interaction between incubation time, plant type and incubation temperature ($F_{4.117}$ = 3.733, $p$ = 0.007)), suggesting that net N mineralization rates were affected by both plant type and temperature. However, post hoc multiple comparisons showed no significant difference ($\alpha$ = 0.05) between the model's effect estimates due to plant type when grouped by temperature except the variable temperature treatment, where model estimates for Rg were significantly smaller than that of FR yet neither was significantly different from HV. K values obtained by linear regression after limiting the model to the incubation time–temperature interaction (coeff $\pm$ SE) were significantly different ($\alpha$ = 0.05), in the order 15 °C (0.364 $\pm$ 0.029 week$^{-1}$) > 2–15 °C (0.275 $\pm$ 0.029 week$^{-1}$) > 2 °C (0.197 $\pm$ 0.029 week$^{-1}$), and an overall adjusted R-squared value of 0.68.

Chemical termination did not have a consistent effect on linear regression estimates between transformed N mineralization data and incubation time but instead affected each plant type differently among temperature treatments (Figure 4). This is supported by statistical analysis in chemically terminated hairy vetch (HV + G) and ryegrass (Rg + G), which shows a significant four-way interaction between incubation time, plant type, temperature and termination method treatments ($F_{2,156}$ = 5.77, $p$ = 0.004). Post hoc multiple comparisons grouped by both plant type and incubation temperature resulted in smaller effect estimates for chemically terminated catch crops but was only significantly so for ryegrass at 15 °C (coeff $\pm$ SE = −0.64 $\pm$ 0.277).

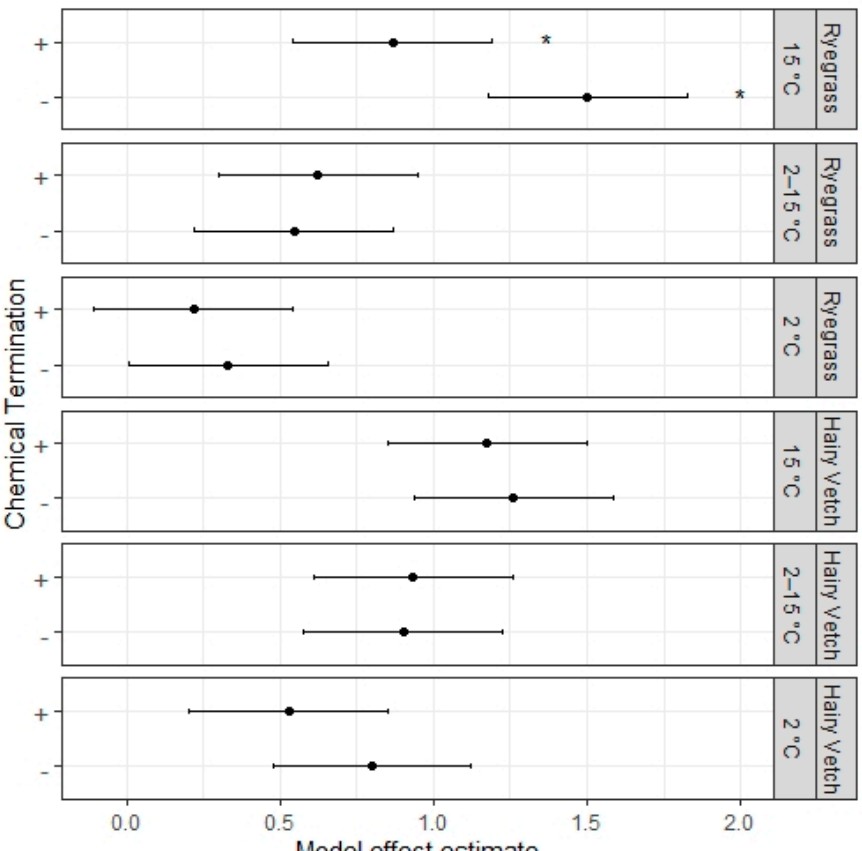

**Figure 4.** Post hoc comparison of linear model estimates, including main effects for net mineralized plant residue N between chemically terminated (+) and untreated (−) hairy vetch and ryegrass. The error bars represent 95% confidence intervals. Asterisks (*) denote a significant difference only within plant–temperature combinations.

### 3.5. Approaches Outside FOK

As the examples in Figure 5 show (see Appendix A for all non-smoothed fitted values), the FOK model fitted well to the experimental $N_{net}$ data. However, it is also visible that this model does not follow the sigmoid shape of some incubation curves (at 2 °C and variable temperature), and both overestimates net N mineralization at early incubation times and underestimates it after week 4. The modified FOK function with exponential lag (exp), Gompertz-type equation (Gomp) and Monod-type equation (Mon), on the other hand, all were capable of fitting the sigmoidal shape of the incubation data from 2 °C and variable incubation temperature treatments, while still providing as good a fit in the 15 °C temperature treatment.

Root-mean-square error (RMSE) values of fitted values against the objective data also indicate that the lagged functions considered provide better fits than first-order kinetics, particularly at low and variable incubation temperatures. In all but one of the incubation curves (chemically terminated hairy vetch at 15 °C), RMSE was equal or lower in for the lagged functions compared to FOK and was lower for all plant types incubated at 2 °C and variable temperature, regardless of chemical termination treatment (Figure 6).

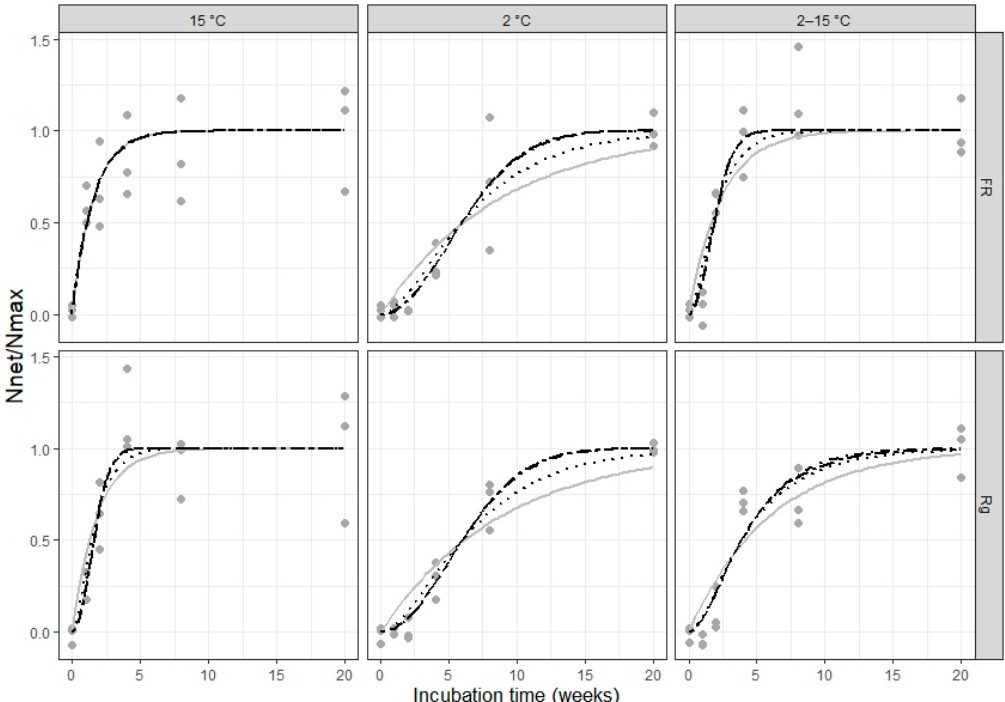

**Figure 5.** Smoothed curves showing results from nonlinear least-squares regression of different functions to $N_{net}$ data, here normalized to $N_{max}$ for ease of view. The gray dots (•) represent the experimental results, and the solid gray lines correspond to fitted values using first-order kinetics (FOK). The black lines represent FOK models modified with a lag term: exponential lag term (dotted), a Gompertz-type equation (dashes), and a Monod-type equation (dot-dash).

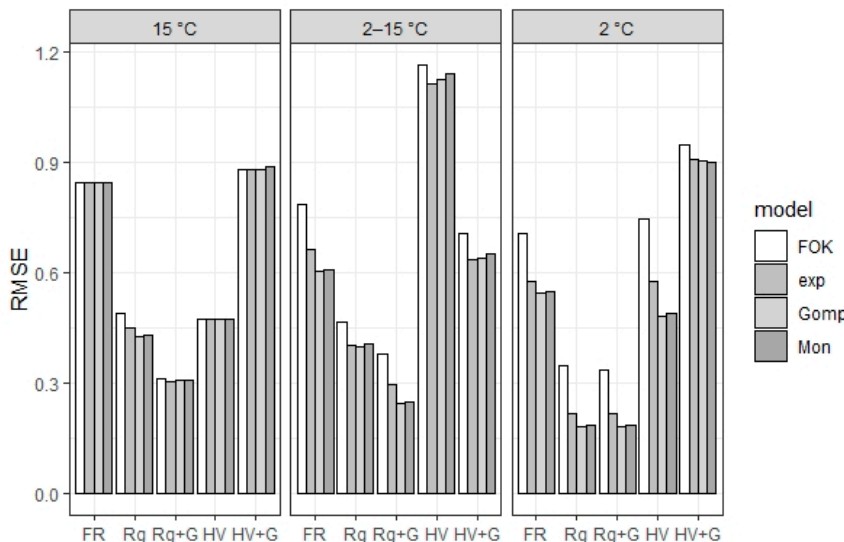

**Figure 6.** Root-mean-square error (RMSE) of different functions fitted to experimental net N mineralization of catch crop residue at different incubation temperatures data by the nonlinear least-squares method.

## 4. Discussion

The timing in the accumulation of ammonium and nitrate in this experiment is interesting for two reasons. First, there is a minimal accumulation of ammonium in the controls, yet there is a steady accumulation of nitrate at all temperatures, which does not decelerate over the five months of incubation. This suggests that a pool of microbial decomposers exists in stable equilibrium with nitrifier microbes, actively mineralizing SOM for the whole

course of the experiment and presumably much longer. Second, there is an observable delay in the onset of ammonium accumulation relative to the addition of fresh plant residue in all plant treatments at 2 °C and variable incubation temperature, which is not present in the corresponding control samples. Corresponding delays in nitrate accumulation relative to the peak in ammonium accumulation, however, are not present in any of the plant treatments regardless of incubation temperature, where the maximum increase in accumulated nitrate occurs at or immediately after the time of peak ammonium accumulation in the sample. This suggests a dormancy and/or low abundance of decomposer microbes, which specifically affects fresh plant residue mineralization, and that this dormancy/absence is quickly overcome at 15 °C but remains for a longer time at lower or variable incubation temperatures. Notably, the initial lag is not influenced by catch crop type (and with it the C:N ratio of the residue), ruling out microbial immobilization of mineral nitrogen as its cause. Thus, we venture that the effect of temperature on the different processes involved in increasing residue-N mineralization activity to its optimum levels (e.g., physiological activation, reproduction and colonization of plant residue surfaces) might be of importance for residue turnover and N mineralization outside of the effect of temperature on mineralization rates once the process is underway.

Statistical tests using linear regression of transformed mineralization data show that the only temperature had a significant effect on net N mineralization after $N_{net}$ was scaled to $N_{max}$ (see Equation (3), $N_{net}$ is divided by $N_{max}$). As would be expected, the FOK rate constant (K) values obtained from regression ignoring plant type were lowest at 2 °C and highest at 15 °C, with the variable temperature treatment between the two. Interestingly, the K value obtained from the 15 °C treatment was very similar to the FOK turnover rate used in APSIM for the carbohydrate fraction of fresh organic matter at the same temperature ($\sim 0.35$ week$^{-1}$) in the soil-N sub-model [40].

Catch crop type, in turn, was the only statistically significant determinant in the maximum net N mineralization fraction ($N_{max}/N_0$), as calculated here. The choice of net N mineralization at 20 weeks of incubation as the maximum net N mineralization from catch crop residue (in the short-term) was based on the observation that there was a minimal additional accumulation of mineral N in the incubation samples between 8 and 20 weeks of incubation in the majority of temperature and plant treatment combinations. This choice was supported by the lack of significant effects from incubation temperature on $N_{max}$, indicating that 20 weeks is likely enough time for the more labile biochemical components in the residue to be mineralized almost completely, even at low temperature. Thus, $N_{net}$ after 20 weeks of incubation is a good estimate of the actual theoretical $N_{max}$ for a fraction of catch crop residue N readily available for mineralization.

Thus, we observe strong independence between the maximum amount of short-term available residue-bound N (a function of plant type) and mineralization rates (a function of soil temperature). Additionally, we observe that the occurrence of initial lag, and likely its duration, is also a function of soil temperature, independent from the taxonomical origin of the plant residue or its content of fast-turnover nitrogen.

Finally, in regards to the effect of chemical termination and its potential for affecting catch crop residue decomposition, we were unable to find a consistent effect on net N mineralization rates across plant types and incubation temperatures or any visible effects on the initial lag at low and variable incubation temperature treatments. This indicates that termination of catch crops with herbicides before incorporation is most likely not an important factor affecting the rate nor the onset of net N mineralization from plant residue. Indeed, other incubation studies have found that chemical termination had a minimal effect on either net N mineralization from incorporated catch crops or the uptake of catch crop N from following crops [41]. The timing between herbicide application and residue incorporation, however, is possibly an important factor for the effect of chemical termination on residue decomposition and N mineralization, as earlier incorporation would increase the risk of introducing active herbicide to the soil environment, and later incorporation could give time for wilting to significantly affect residue quality. We, therefore, suggest that more

research involving the timing between herbicide application and residue incorporation is needed to properly establish the effects of chemical termination on N mineralization.

We consider it important to acknowledge the difficulties in drawing conclusions from experimental incubation data by fitting functions based on different assumptions over the underlying processes to a relatively small set of experimental results and that our assertions would greatly benefit from being tested on other datasets. Comparing results from independent incubation studies, however, involves difficulties of its own, particularly regarding preparation techniques and the early stages of incubation. Preliminary steps, such as pre-incubation, drying and rewetting of soil, drying and grinding of plant residue, among others, are generally acknowledged to affect microbial decomposer activity yet vary greatly between incubation experiments. For instance, Van Schöll et al. [9] incubated fresh winter rye shoots in soil that had been in fallow during autumn and winter, specifically avoiding drying and rewetting not to alter the soil's microbial composition. Baijukya et al. [42] incubated decomposing leaf and stem residues from different crops also using fresh soil but mixing it with an equal mass of acid-washed sand. On the other hand, Bending et al. [43] measured N mineralization from a variety of fresh crop and catch crop residues incubated in rewetted air-dried soil following pre-incubation at 20 °C for five days, although they did not write the rationale behind this pre-incubation or its parameters. Meanwhile, Jensen et al. [44] incubated oven-dried clippings from a wide variety of crops and cover crops in soil that had been pre-incubated with a mixture of red clover and timothy for 53 days explicitly in order to "boost a rich and potent decomposer community and thus alleviate possible limitations due to poor colonization of fibrous material". These differences in protocol make it difficult to compare effects, such as the initial lag discussed here, as drying and rewetting, pre-incubation, and the addition of acid-washed sand or extra fresh plant residue almost certainly have a strong effect on the number, diversity and physiological status of soil microbes entering the incubation procedure. Likewise, decisions on the fragment size in the added plant residue, and whether it is dry or fresh, homogeneously mixed or added in a layer, etc., likely influence the process of decomposer colonization of the fragment surfaces. We finally argue that some degree of lag in net N mineralization is appreciable in the reported results of all, but the latter of the aforementioned studies, yet there is almost complete uncertainty as to the cause and significance of said lag in each individual case.

Clearly, much work is still required to establish the importance of initial delays in residue decomposition, both in terms of its effect on soil fertility and N leaching, as well as the conditions leading to it. In a modeling context, there would be particular difficulties inherent to disentangling turnover rate parameters from initial lag parameters, as well as determining the temperature, water content and pH functions that would affect each. However, we were able to decrease the error associated with fitting an FOK function to low-temperature results without increasing the error at higher temperatures by adding relatively simple lag terms to a widely used and robust mathematical and experimental approach. We suggest, therefore, that if found to be sufficiently relevant, it should be possible to introduce initial lag effects into cropping models without requiring a major restructuring of existing SOM turnover sub-models.

## 5. Conclusions

Our results show that the onset of catch crop residue N mineralization is delayed at low (2 °C) and variable temperature (variation between 2 °C and 15 °C every two or three days), but nitrification of newly produced ammonium occurs promptly at peak soil ammonium content regardless of temperature treatment. Furthermore, we observed no significant differences in the final accumulation of mineral N due to incubation temperature and no consistently significant differences in overall net N mineralization or in the onset of mineral N accumulation in soil from chemical termination. Incubation temperature did affect overall N mineralization rates as would be expected, causing a decrease in fitted first-order kinetics rate constants in the order 15 °C > variable temperature > 2 °C.

This leads us to believe that an important component in the effect of low temperature in net N mineralization comes from a delay in microbial decomposer activity shortly after residue incorporation, likely involving colonization of the added residue. Given that the observed delay, or lag, in net N mineralization was present in all plant treatments equally at low and variable incubation temperature, it is not likely caused by microbial N immobilization.

Thus, we propose that low and variable incubation temperature affects the activity of decomposer soil microbes independently from plant residue type, changing the mode of net catch crop residue N mineralization from exponential to sigmoidal. This is supported by the results from fitting exponential functions modified to initial lag to our observations, which yielded consistently better goodness-of-fit statistics than a function derived from first-order kinetics alone.

Finally, we propose that with simple but carefully chosen lag functions, it is possible for cropping models to account for initial lags in residue N decomposition at low temperatures without abandoning the current use of first-order kinetic equations for most soil turnover processes. However, the effectiveness of this idea is tied to the development of experimental procedures that can differentiate the effects of temperature on the initial lag from other factors.

**Author Contributions:** Conceptualization, J.F.M.-V. and I.V.; methodology, J.F.M.-V. and I.V.; validation, I.V.; formal analysis, J.F.M.-V.; investigation, J.F.M.-V. and I.V.; data curation, J.F.M.-V.; writing—original draft preparation, J.F.M.-V.; writing—review and editing, I.V.; visualization, J.F.M.-V.; supervision, I.V.; project administration, I.V.; funding acquisition, I.V. All authors have read and agreed to the published version of the manuscript.

**Funding:** This research was funded by the Ministry of Environment and Food of Denmark under GUDP project "Sat-N".

**Data Availability Statement:** The data in this study are openly available in FigShare at doi:10.6084/m9.figshare.14339579.

**Conflicts of Interest:** The authors declare no conflict of interest.

## Appendix A

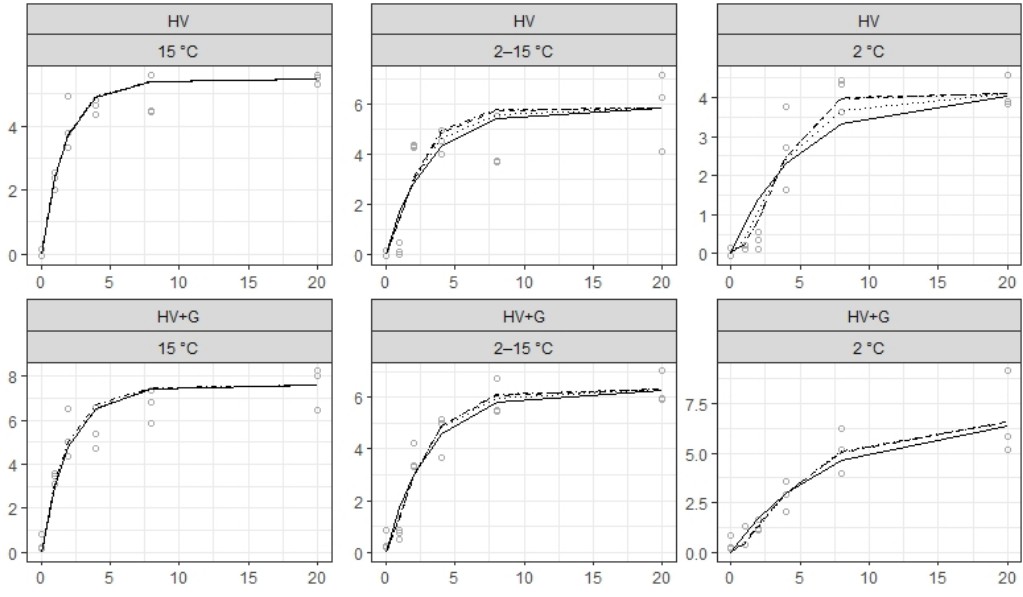

**Figure A1.** *Cont*.

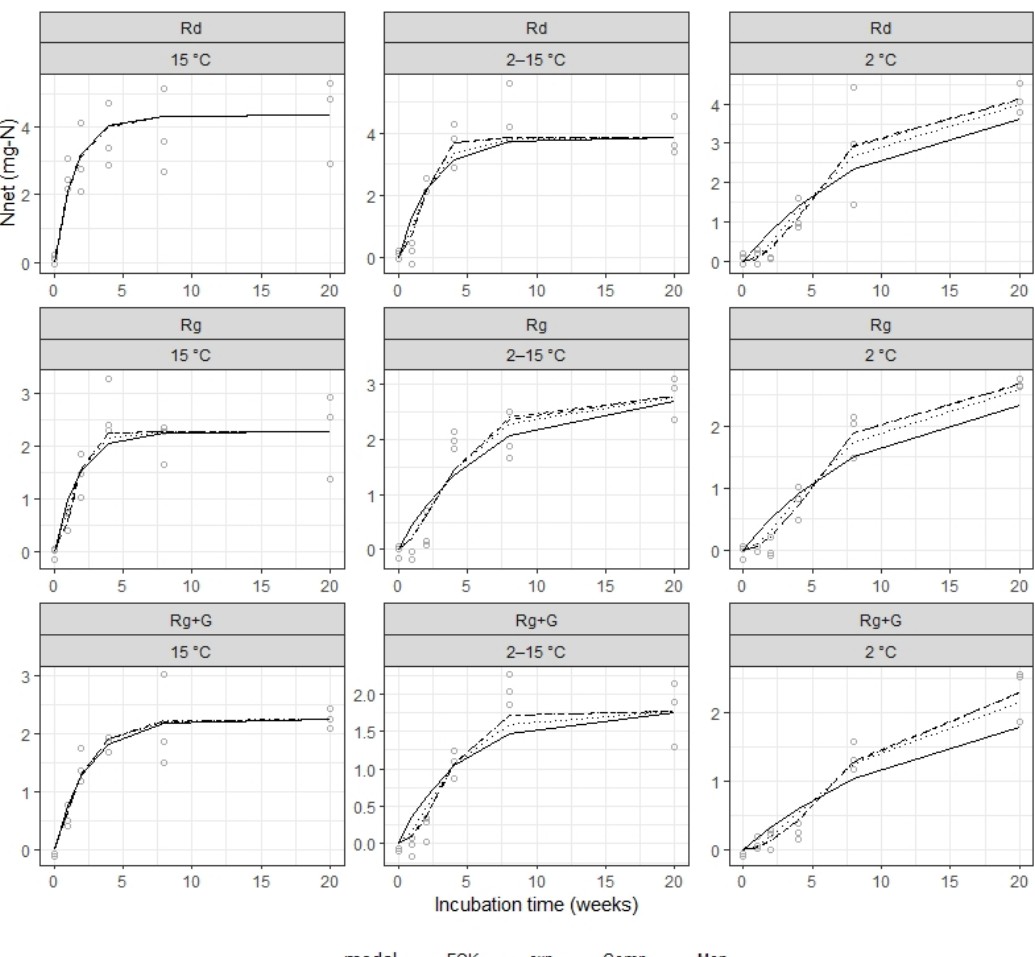

**Figure A1.** $N_{net}$ (mg-N) values fitted by nonlinear least-squares regression using a first-order kinetics model (FOK), an FOK model with an exponential lag term (ex), a Gompertz-type equation (Gom), and a Monod-type equation (Mon). The gray circles (◯) represent the experimental objective data. All amounts are reported as total weight (in mg of N) per soil core (~250 mg DW).

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
