# Peer review of "Exploring Temperature-Related Effects in Catch Crop Net N Mineralization Outside of First-Order Kinetics"

_nitrogen, doi:10.3390/nitrogen2020008_

Round 1

Reviewer 1 Report

The contribution is devoted to important questions of fate of N from catch crops (CC) biomass. CC are taken as the main solution for reduction of N leaching during autumn-early spring seasons, however it must be considered within plant-soil-atmosphere complex of interacting factors. In many farms the CC residues are incorporated early at autumn resulting in possible risk of N leaching.

The experiment was carefully prepared and performed, detailed description of methods and materials enable repeatability and comparison with data of other authors (duly discussed by the authors). Among others, the authors considered possible N losses by denitrification. The experimental data were thoroughly presented and analysed.

The contribution is suitable for publication; I have only a few comments.

l. 9-13 Dividing the long sentence (from quantifying ....?) would improve readability 

l. 18 A bit unclear, should be specified that “initially-added N” in CC material/residues)

l. 27 from agriculture?

l. 116 m3 – upper index

l. 130 Please, add age or dates of plants sowing and sampling, possibly the stage of development, as it may affect the C:N and decomposition rate (e.g. in Brassicas)

Fig.1. 2, 7 Standard units should be in the format mg/weight unit or per cylinder etc. (you also use “concentration”  in the text). I understand the weight is all the same, but consider the modification, or give, at least, the unit (weight of soil) in captions. Further, in Fig. 7 Nnet (mg-N)  x Fig. 1 and 2.

Also, as you do not use legends at the graphs, you should show the symbols in captions. 

l. 341 NO3 – lower index

l.356 Two dots

l. 368 Is the use of two forms of units right (Mg/m3 x week-1)

l.482 Van Scholl et al. ? also l. 485 x l. 490

References

The use of quotation marks is unusual

There different styles of journal names, non-, and abbreviated, and format – year at last or last but one positions – should be unified

Ref. 17, 19, 20 and other? 

Author Response

The authors thank Reviewer 1 for their positive comments, and for their prompt and attentive review. Following are responses to the specific comments made:

l. 9-13 Dividing the long sentence (from quantifying ....?) would improve readability 

Fixed

l. 18 A bit unclear, should be specified that “initially-added N” in CC material/residues)

Fixed

l. 27 from agriculture?

Fixed. Now reads: "In agriculture, catch crops are a widespread and effective method for reducing post-harvest soil mineral nitrogen, and thereby leaching of nitrogen after harvest and during the autumn-winter season."

l. 116 m3 – upper index

Fixed

l. 130 Please, add age or dates of plants sowing and sampling, possibly the stage of development, as it may affect the C:N and decomposition rate (e.g. in Brassicas)

Fixed. The text now states that the catch crops were sown in autumn, overwintered in the field and hadn't yet begun flowering in spring, as well as the presence of large taproots in fodder radish.

Fig.1. 2, 7 Standard units should be in the format mg/weight unit or per cylinder etc. (you also use “concentration”  in the text). I understand the weight is all the same, but consider the modification, or give, at least, the unit (weight of soil) in captions. Further, in Fig. 7 Nnet (mg-N)  x Fig. 1 and 2.

Fixed. The relevant figure captions now include the text "All amounts are reported as total weight (in mg of N) per soil core (~250 mg DW)", and the incorrect use of "concentration" for soil solute content has been corrected.

Also, as you do not use legends at the graphs, you should show the symbols in captions. 

Fixed.

l. 341 NO3 – lower index

Fixed.

l.356 Two dots

Fixed.

l. 368 Is the use of two forms of units right (Mg/m3 x week-1)

Fixed. The use of dash was avoided in favor of negative power.

l.482 Van Scholl et al. ? also l. 485 x l. 490

Fixed.

References

The use of quotation marks is unusual

Fixed.

There different styles of journal names, non-, and abbreviated, and format – year at last or last but one positions – should be unified

Fixed.

Ref. 17, 19, 20 and other? 

Fixed.

Reviewer 2 Report

Title:

Exploring temperature-related effects in catch crop net N mineralization outside of first-order kinetics

Recommendation:

Minor revision.

Comments:

This study investigated the temperature-related effects in catch crop net N mineralization outside of first-order kinetics. The subject is relevant and consistent with the aims and scopes of the journal. In my opinion, the manuscript is interesting and produces new knowledge on the application of catch crop net N mineralization. Some comments and suggestions are offered below with the intent to assist the author in improving the manuscript.

  1. In the research design, the researchers selected low temperature (2°C, 5°C, and 2-15 °C) as the analysis conditions. Does this temperature represent the local four-season temperature of the research site? Since the physiological activities of microorganisms and plants are different at different temperatures, the rationality of the experimental design is very important. Please provide more information for this part.
  2. This study chose three catch crops (hairy vetch, fodder radish and ryegrass). Please briefly explain the reasons for choosing these three plants in the manuscript (is there any relevant references?), and briefly describe the growth cycle and characteristics of these three plants.
  3. There are some typo in the manuscript. For example, the numbers of NH4-N and NO3-N in Figure 1 need to be subscripted. Please re-check the manuscript carefully and correct them.
